# A Retrospective Study on Adoptive Parenthood in the First Year after the Adoption: The Role of Parents’ Attachment and Empathy on Communicative Openness

**DOI:** 10.3390/healthcare11243128

**Published:** 2023-12-08

**Authors:** Laura Gorla, Chiara Fusco, Alessandra Santona

**Affiliations:** Department of Psychology, University of Milano-Bicocca, 20126 Milano, Italy; c.fusco7@campus.unimib.it (C.F.); alessandra.santona@unimib.it (A.S.)

**Keywords:** adoption, parental attachment, family communication, adoptive families

## Abstract

Communicative openness (CO) defines the willingness of parents and children to explore the significance of adoption. Especially in the first year of adoption, CO could be challenging for adoptive parents, who are influenced by personal characteristics. Using a retrospective assessment, we investigated parents’ communicative experiences in the first year of adoption and whether these are affected by romantic attachment and empathy. In the study, 290 adoptive parents (females = 73%, mean age 50 years) filled (a) an ad hoc questionnaire for CO, (b) Experiences in Close Relationships-Revised (ECR-R) for attachment, and the (c) Interpersonal Reactivity Index (IRI) for empathy. During the first year, most parents reported difficulties in controlling their emotions and understanding their children’s emotions. Parents with an avoidant attachment and personal distress in empathy were more likely to feel fatigued in sharing and controlling personal feelings and understanding their children’s feelings. Open adoption-related communication is a complex and challenging process for adoptive parents, which can be facilitated or not by individual characteristics such as avoidant attachment and personal distress in emotional situations. These results could help develop psychological interventions targeting adoptive parents during the first year after the child enters the family system.

## 1. Introduction

### 1.1. Communicative Openness about Adoption

Communicative openness (CO) refers to a complex and dynamic process involving adoption-related communications within the adoptive family. Several scholars [1,2] have defined CO as a continuum of communications, indicating the willingness of parents and children to explore the significance of adoption in their lives and share it with significant others in their social networks [2,3]. CO evolves throughout various stages of the family life cycle [4,5]. Indeed, it is characterized by various interactive family processes, including emotional attunement, empathy, expressing emotions, and co-constructing and sharing meanings related to adoption [2,5].

In adoptive families, dialogue is crucial. It has been argued that adoptive families represent an example of “discourse-dependent families” [6]. To shape their sense of family identity, adoptive families emphasize communication and construct narratives not only to establish and reaffirm their familial bonds, but also to continually redefine their family dynamics [5,7,8]. Alongside that, an open dialogue in the adoptive family is essential in supporting the child’s development of an adoptive identity that integrates the adoptee’s past and present experiences and adoptive status [5,6]. Effective communication can help the child understand and make sense of their adoptive experience and foster the affective bond between adoptive parents and the child [9,10,11]. Therefore, research emphasized the importance of encouraging open communication between family members from the initial moments of the child’s life within the adoptive family [1,12].

However, recent research by Soares et al. [13] points out that CO about adoption can be a challenging process for adoptive parents and adoptees. It has been identified as the main obstacle reported by children during the post-adoption period [4,14,15]. Specifically, parents often find it challenging to handle their children’s questions and behaviors related to adoption, as they involve sensitive aspects and require the ability to understand their children’s needs and emotions [14,16,17].

### 1.2. The First Year after Adoption

The first year after adoption represents a critical period for the adjustment of the adoptive family. Indeed, the transition to adoptive parenthood is characterized by a significant impact on adoptive parents, both from physical and emotional perspectives [18,19]. First, prospective parents have to deal with a waiting period between their certification of eligibility for adoption and matching with a child. This period can vary but is usually quite long and can be experienced as significantly distressing by couples [18]. Thereafter, with the adoptee’s entry into the family, adoptive parents have to adjust to their new parental role and duties, entitling themselves to be parents of that specific child, overcoming the absence of a genetic link, and committing themselves to the child’s positive inclusion within the broader family system and into the social environment in which the family is placed [7,20].

Moreover, the first year after the adoption is crucial for developing the attachment bond between the child and the adoptive parents [21]. Previous studies pointed out that one year after adoption, adopted children who had previously spent time in institutions seem to exhibit higher rates of insecure and disorganized attachment patterns compared to the normative population and those adopted within their first year of life [22,23]. However, the rates were still lower than those found in studies involving children currently placed in institutions [24], indicating the protective effect of the adoption in modifying the attachment patterns of previously institutionalized children [21].

In addition, the first year after the child’s entry into the adoptive family is crucial, as the adoptee starts to build his or her adoptive identity, where the primary developmental task is to integrate the dual belonging of the child to two families (the birth and the adoptive families). Within this process, literature has shown the CO role in promoting positive developmental outcomes for adopted children [3,11] during and beyond the first year after adoption. Communicating about adoption seems to lead to greater satisfaction with children’s adoptive status [25], fostering stronger bonds within the adoptive family [26,27], and contributing to a coherent and integrated adoptive identity [11,28]. Furthermore, CO helps adoptees develop positive views of their biological parents and the adoption history, reducing feelings of rejection and loss [2,3].

### 1.3. Challenges in Communication: Factors Influencing CO

Several factors have been observed to influence communication within the adoptive family. Among these, research has shed light on gender-related differences in parent–child communication [29,30]: daughters, regardless of their biological or adoptive status, often reported better communication with their parents than sons did. In addition, their offspring often perceived mothers as more available to communicate with than fathers. These findings imply that gender-related communication differences extend beyond biological parent–child relationships.

Another factor potentially impacting parent–child communication is the child’s age, which may influence their understanding and curiosity about their origins [14]. Adolescence is widely known as an identity-defining life stage that, for adopted people, can occur with specific complexities, influencing the timing, content, and ways in which the entire family communicates about adoption [1,31]. Due to the cognitive development associated with adolescence, adopted adolescents are more able to understand the possible motivations that led to their adoption, redefining their identity through a dialectical communicative process involving adoptive parents [1].

Moreover, either the lack of information or having to manage how to communicate complex and harsh information about the adopted child’s past could represent a difficulty for the parents’ handling of open communication [32,33,34]. Psychosocial literature in this field has highlighted the relevance of parents’ honesty and transparency with the child, tailoring their storytelling to match the adoptee’s developmental phase, inquiries, or observations [33]. However, several challenges may surround open communication about adoption and may be related to various factors linked to adoptive parents’ personal histories. These difficulties potentially encompass experiences like the grief and loss associated with infertility, the complexities and duration of the adoption process itself, and the emotions evoked by the origins of their adopted child [7]. These emotions could include fear, insecurity, or a sense of rejection [3,35]. Moreover, a recent study by Alves et al. [35] highlighted that the most severe parental difficulties related to the parent–child relationship emerged during the first year after adoptive placement. Despite their relevance for family functioning and communication, little is known about the difficulties that adoptive parents perceive in communicating about adoption, particularly during the first period after the child enters the family system.

### 1.4. Adoptive Parents’ Attachment and Open Communication

Given its importance for the psychological adjustment of adopted children and adolescents, open communication has been emphasized as requiring active parental involvement [12]. Research indicates that adoptive parents play a central role in creating a supportive environment that fosters CO in their children [2,36]. Parents’ attitudes and behaviors are believed to establish the initial context that encourages openness in the child, either nurturing or stifling the child’s curiosity [17,26].

Among these parents’ characteristics, parental attachment patterns may serve as a relevant framework for understanding parenting dynamics and family functioning related to CO about adoption. Attachment patterns have been classified into two underlying dimensions, defined as attachment anxiety and avoidance. Attachment anxiety is characterized by the fear of abandonment in close relationships, vulnerability to rejection by significant others, and the use of hyperactivating strategies in dealing with attachment issues [37,38]. Attachment avoidance describes individuals who tend to be self-reliant, feel discomfort with emotional closeness and intimacy, and employ deactivating defensive strategies in attachment-related experiences [37,38].

Numerous studies have demonstrated the significant impact of attachment models on various aspects of adults’ interpersonal behavior [3,37,38,39,40]. Attachment patterns contribute to influencing how individuals perceive and interact with others, their expectations of others’ behaviors, and their overall approach to close relationships [38].

Research has emphasized a bidirectional link between attachment and communicative systems [41,42]. Individual attachment styles and states of mind influence communication patterns; at the same time, early communicative experiences play a role in developing secure attachment relationships [42]. Specifically, individuals with a secure attachment style tend to engage in behaviors that foster closeness, such as self-disclosure, nonverbal intimacy, and affectionate communication, while those with high attachment anxiety may exhibit excessive or inappropriate self-disclosure [43], and individuals with high attachment avoidance are more likely to withhold information from significant others [44,45]. Moreover, studies exploring the influence of parental attachment styles on parent–child communication have highlighted the connection between interaction patterns with children and parents’ attachment states of mind [46,47]. In addition, adoptive parents’ secure attachment style was crucial for promoting positive changes in the attachment patterns of the adopted child [9,23,48].

### 1.5. Parents’ Empathy and Open Communication

Parents’ empathy is key when considering the family’s functioning and communication processes [2]. Psychological literature conceptualizes empathy as the capacity to imagine, experience, and understand what another person is feeling. Typically explained as a two-component model, it is characterized by an affective and cognitive dimensions. The first is represented by compassion, interest, or personal distress when exposed to others’ emotional states, whereas perspective-taking abilities reveal the empathy-related cognitive dimension [49]. Empathy has also been divided into three dimensions: perspective-taking, empathic concern, and empathic distress. The first refers to a cognitive tendency to recognize emotions in others and consider their point of view, while the second defines a tendency to sympathize with others’ feelings. Finally, the third aspect reflects the tendency to experience personal distress when exposed to others’ emotions [50].

Empathy is a very complex psychological process, as it is both an experiential emotional state and an individual trait [51], and has been connected to individuals’ attachment representations [52]. Within this framework, parental empathy has been defined as the empathic processes specific to the parent–child relationship, being the parent’s ability to correctly recognize, comprehend, and convey empathic responses toward the child’s emotions [53]. Several studies highlighted that this aspect is related to positive parenting behaviors [50,51,53,54].

Focusing on adoptive families, the level of openness and comfort in discussing adoption issues within the family and the child’s perception of parents’ empathy and respect play a significant role during all children’s developmental stages [33]. Open communication within the adoptive family seems closely tied to parents’ empathy and adoptees’ overall satisfaction with the adoption status, as it may influence their drive to seek an understanding of their history and origins [2,33,55].

First, Kirk [56] introduced the Shared Fate Theory, which underscores the important role adoptive parents play in promoting their children’s positive feelings and emotions about their origins. Among the factors involved in this process, this theory emphasizes the significance of empathy, alongside the distinction between adoptive and nonadoptive parenthood, and the role of effective communication within the adoptive family. From this perspective, adequate empathic abilities may let adoptive parents better perceive and understand their children’s characteristics and experiences related to their adoption story and previous background [55]. In this context, adoptive parents with high levels of confidence in managing their own emotions concerning their child’s origins are more likely to anticipate and embrace their child’s needs, promoting effective and open communication. Remaining neutral or indifferent due to uncertainty about what to say or downplaying the significance of the child’s origins can create a sense of detachment between adoptive parents and their child [33,57].

The psychological literature exploring open communication within adoptive families has emphasized the importance of parents’ empathy. A recent qualitative study [33] found that interviewed adoptive parents showed a strong empathy toward their children’s need to explore and gain deeper insights into their origins, regardless of age. They acknowledged that facilitating this process was pivotal in shaping their family’s narrative communication, offering lasting advantages to their children’s identity development and life story. In addition, several studies highlighted that empathic parents are more likely to communicate openly and cooperatively with their adopted offspring [33,54] and pointed out the importance of providing psychological interventions focused on empathy within the family [58]. Following the recent and broad literature relating empathy and communication within the adoptive families, the current study focused on the specific interplay between parents’ empathy and parents’ personal experiences, and difficulties in open communication regarding adoption with their children. To our knowledge, parents’ difficulties in the CO process have been investigated in few studies, especially with quite a large sample. Thus, more psychological research exploring the potential link between adoptive parents’ empathic abilities and CO about adoption experiences seems needed.

### 1.6. The Present Study

Openly communicating about adoption has been considered challenging for adoptive parents [13], although it represents a key factor for the entire family’s well-being [11]. Adoptive parents could perceive an open dialogue about adoption as particularly harder in the first year of adoption, where the child and parents need time to know each other and create their family identity.

To the best of our knowledge, there is a gap in the literature regarding the connection between the first year’s communication about adoption and parents’ characteristics of attachment and empathy. The current study aimed to fill this gap by focusing on parents’ experiences in open communication about adoption in the first year of the adoption. Our study investigated the adoptive parents’ experiences while communicating with their children, their difficulties, and the role of parents’ attachment and empathy in the first year’s communication.

In particular, the first part of the current study aimed to explore open communication in the first year by collecting a retrospective description of what happened during the first year of the adoption, as parents were asked to tell how much they communicated their emotions while communicating and their difficulties in talking about specific themes. Considering the exploratory nature of this part of the study, we did not have specific hypotheses for CO during the first year.

However, based on previous literature on this topic, we expected that parents’ attachment and empathy would have been related to their experiences while communicating about adoption-related themes with their children.

Specifically, attachment modulates individuals’ strategies for regulating interpersonal closeness in intimate relationships. Individuals with anxious attachment orientations are usually concerned about being abandoned by significant others, thus hyperactivating proximity-seeking strategies. Similarly, individuals with avoidant attachment tend to face difficulties engaging in close and intimate relationships and subsequently avoid emotional closeness. Moving from these assumptions, we expected that parents showing high levels of attachment anxiety and avoidance would experience more difficulties in emotionally and relationally demanding situations, such as adoption-related communication.

Regarding empathy, we hypothesized that parents’ empathic abilities, such as perspective-taking and empathic concern toward the child, would represent a protective factor connected to CO. Conversely, we hypothesized that parents’ distress when exposed to adopted children’s adverse emotions would have been connected to parental difficulties in CO.

Three main hypotheses guided the current study:


(a)Parents with high levels of attachment avoidance and anxiety experienced difficulties while openly communicating with their children;(b)Parents showing perspective-taking abilities and empathic concern toward their children perceived open communication as not difficult;(c)Parents with personal distress in empathizing with their children in emotional situations experienced difficulties discussing adoption with their children.


## 2. Materials and Methods

### 2.1. Participants

Our sample comprised 290 adoptive parents (females = 210, 73%), with a mean age of 50 years (SD = 6.9). They were mainly married (276, 97%) and living in Northern Italy (178, 64%). Most participants had a bachelor’s or master’s degree (54%, N = 153), and a middle socioeconomic level (45%, N = 127). Adoptive parents and adopted children have lived together for a mean of 10.2 years (SD = 5.18).

Adopted children were primarily males (63%, N = 176), aged 13 years (SD = 5.2), and adopted at a mean age of 4 years (SD = 3.0). Most of them were internationally adopted (73%, N = 208) and were born mainly in Columbia (13%, N = 38), Ethiopia (7%, N = 21), and Russia (6%, N = 17). Before being adopted, most adopted children lived outside the birth families (e.g., in hospitals or foster care; 90%, N = 257) and never suffered from health issues (61%, N = 172). Finally, most of our participants were not followed by adoption agencies (74%, N = 210).

### 2.2. Procedure

We collected data in collaboration with several adoption agencies all over Italy. The research assistants contacted the adoption agencies, explained the research and asked them to collaborate. If the adoption agencies were interested, they sponsored the research using a newsletter or sent adoptive parents an email with a description of the study and a Qualtrics link to the research protocol. Participants filled questionnaires using the Qualtrics platform. Data collection started in January 2022 and ended in June 2023.

We proposed our study only to adoptive parents who adopted at least one year before the research protocol administration and had children aged between six and eighteen years. We followed the provisions of the Italian law 196/2003 to collect the participants’ consent to participate in the research. Before starting, the participants read a brief explanation about the content and purpose of the study. The Research Ethics Committee of the Psychology Department of Milano-Bicocca University previously approved the research project.

### 2.3. Measures

Our participants filled the following instruments:a.Communicative openness within the family in the first year after adoption

To explore what happened in the family’s first year of life together, we created an ad hoc questionnaire by asking parents the following:-How frequent was the communication about adoption-related themes in the first year;-What was the theme most frequently discussed within the family and how (e.g., using adoption-related books, pictures, and documents from the adoption agencies…);-Which emotions were present during the communication;-If and why some information about the child’s past life had been hidden or not shared within the family;-If this information was at some point disclosed, we investigated when, how, and why parents shared it with their children;-What kind of difficulties did parents encounter in talking about adoption.

In the current study, we focused mostly on parents’ difficulties discussing adoption-related aspects in the first year after adoption. In particular, we asked parents to express how frequently they encountered difficulties sharing personal feelings, controlling personal emotions, understanding children’s emotions, and handling children’s questions (1 = Never, 5 = Always). These difficulties were selected after a long and detailed literature analysis, considering the difficulties more often expressed by parents in after-adoption follow-ups and interventions.

We also created binary variables for parents’ difficulties during CO by dividing scores of “yes” if participants reported they experienced difficulties as “sometimes”, “often”, and “always”; and “no” if they answered they “never” and “rarely” had difficulties in CO.

As single items expressed difficulties, no Cronbach’s alpha coefficients were calculated.

b.Experiences in Close Relationships-Revised (ECR-R)

We used the Italian version of the Experiences in Close Relationships-Revised (ECR-R) [59,60,61] to assess the behaviors and feelings connected to attachment of the romantic couple. The questionnaire is composed of 36 items having a range from 1 (“strongly disagree”) to 7 (“strongly agree”), with higher scores showing a higher endorsement of the construct. ECR-R is characterized by two subscales: avoidance of intimacy and anxiety about abandonment. The first subscale measures worries connected to sharing emotional closeness (e.g., “I prefer not to show my partner how I am really feeling inside”), while the second one indicates the levels of preoccupation with the relationship and the need for intimacy (e.g., “I am afraid that I will lose my partner’s love”). In our sample, Cronbach’s alpha coefficient was 0.76 for avoidance and 0.86 for anxiety, showing good reliability.

c.Interpersonal Reactivity Index (IRI)

We used the Italian version of the Interpersonal Reactivity Index (IRI) [62,63] and adapted the instrument to parent–child relationships, following previous literature [50]. The instrument measures empathic responsiveness, considering empathy as a set of cognitive and emotional aspects. IRI is a 28-item scale composed of four subscales: the Fantasy Scale (FS), the Perspective-Taking Scale (PT), the Empathic Concern Scale (EC), and the Personal Distress Scale (PD). As we were interested in empathic aspects connected with attachment, we eliminated the Fantasy Scale in the current study.

The Perspective-Taking Scale (e.g., “I sometimes find it difficult to see things from my child’s point of view”) evaluates whether parents try to adopt their children’s points of view. The Empathic Concern Scale (e.g., “I often have tender, concerned feelings for my child”) assesses parents’ feelings of care and compassion toward their children. Finally, the Personal Distress Scale (e.g., “When my child is in emergency situations, I feel apprehensive and ill-at-ease”) measures the extent parents experience anxiety and worries when exposed to the negative experiences of their children.

In our sample, Cronbach’s alpha coefficient was adequate for PT (0.83) and PD (0.67), rather low for EC (0.56).

### 2.4. Analysis Plan

All the analyses were performed using Jamovi 2.3.28 statistical software (The Jamovi project, Sydney, Australia). Firstly, we created binary variables by categorizing parents’ difficulties during CO of “yes” if they reported they experienced difficulties as “sometimes”, “often”, and “always”; and “no” if they answered they “never” and “rarely” had difficulties in CO. To determine the minimum sample size required to detect a significant effect, we computed an a priori power analysis using G*Power 3.1.9.7. [61]. For Point-biserial correlations, with an effect size of 0.3, an alpha level of 0.05, and a statistical power of 0.95, we found the minimum number of participants required to be 134. For logistic regressions, with an odds ratio of 1.5, an alpha level of 0.05, and a statistical power of 0.80, we found the minimum number of participants required to be 215.

We conducted descriptive statistics, Point-biserial correlations among research variables, and logistic regressions to explore the relations between parents’ characteristics (i.e., attachment and empathy) and having or not experienced difficulties in the CO in the first year.

## 3. Results

### 3.1. Descriptive Analyses

#### 3.1.1. First-Year Communication, Parents’ Attachment, and Empathy

Parents reported that they frequently communicated about adoption-related themes with their children during the first year after the adoption, and described themselves and their children as calm but sad during these conversations.

When asked about the contents discussed with their children, most parents (68%, N = 192) reported sharing only some information about their children’s past life as they were primarily scared of emotionally hurting the child. Alongside that, most adoptive parents (58%, N = 167) considered it difficult to share information about the child’s biological mother.

Nevertheless, among the parents who did not disclose every information about their child’s past during the first year after the adoption, many (68%, N = 126) did it in the following years, especially when the child reached an adequate age to understand and emotionally handle the information.

Table 1 reports the difficulties, doubts, and fatigue parents experienced while openly communicating about adoption-related themes in the first year after adoption. In particular, the number and percentage of parents expressing whether or not they have experienced difficulties are reported. Table 2 reports the means of parents’ attachment and empathy.

#### 3.1.2. Correlations

Table 3 reports the correlations between parents’ difficulties in CO and parents’ attachment and empathy. We discovered a significant correlation between parental avoidant attachment and difficulties in sharing personal feelings and understanding children’s emotions. High levels of avoidant attachment are related to more difficulties in these two CO aspects. Conversely, we did not find significant correlations between parents’ anxious attachment, perspective-taking, and empathic concern and all the difficulties evaluated.

We also found a significant correlation between personal distress and difficulties in sharing personal feelings, controlling personal emotions, and handling children’s questions. High levels of parents’ emotional anxiety and worries when exposed to their children’s negative experiences are significantly related to difficulties in the communicative openness process.

### 3.2. Logistic Regressions

We performed logistic regressions using parental difficulties (i.e., sharing feelings, controlling emotions, understanding emotions, and handling questions) as dependent variables, and parents’ attachment and empathy as independent ones. All the regressions included parents’ gender and age as the control variables. We discovered only a significant relation between parents’ gender and difficulties in understanding children’s emotions (β (SE) = −1.09 (0.331), Z = −3.29, *p* < 0.001, OR = 0.335).

Our results showed that avoidant attachment, perspective-taking abilities, and personal distress were related to difficulties sharing personal feelings. Table 4 shows that high levels of avoidant attachment (OR = 1.03, β = 0.03, *p* = 0.003) and personal distress (OR = 2.20, β = 0.78, *p* = 0.006) increase the odds of reporting more difficulties sharing personal feelings during CO in the first year after the adoption. Conversely, parents with perspective-taking abilities are less likely to find it difficult to share personal feelings with their children during CO (OR = 0.63, β = −0.46, *p* = 0.036).

Our results also showed a significant relationship between parents’ distress and difficulties controlling emotions while communicating. Perceiving anxiety and worries when exposed to negative experiences of adopted children increases the odds of struggling to control emotions while openly discussing adoption-related themes (OR = 2.09, β = 0.73, *p* = 0.004). There were no significant relations between parents’ attachment and parents’ difficulties in controlling emotions.

Moreover, parents’ attachment and empathy did not relate to their difficulties in understanding children’s emotions, but parental personal distress related to difficulties handling children’s questions. Indeed, parents who reported anxiety and worries when exposed to negative experiences of their children are more likely to struggle to handle children’s questions during open conversations about adoption (OR = 1.95, β = 0.69, *p* = 0.012).

Table 4 and Table 5 report the goodness of fit measures and results. The chi-squared tests revealed a statistically significant relationship between the predictor variables and the outcomes. This suggests evidence of an association between the predictors and the outcomes. However, it is worth noting that the R-squared values, which indicate the proportion of variance explained by the models, were relatively low. This implies that the current models may have limited predictive power. Exploring additional variables or refining existing ones would be beneficial to enhance the models’ performance.

## 4. Discussion

The current study aimed to explore the experiences parents had during the first year of the adoption while communicating with their children about adoption and to test whether parental attachment style and empathy were related to the communicative openness process. The focus on the first year of adoption was decided as the first year after the child’s entry into the adoptive family is crucial for building an adoptive identity through open and emotionally attuned communication [3,11].

Using a retrospective design, we asked parents to describe how they communicated the first year after the child entered the family system, and they recounted a frequent, calm, and emotionally sad dialogue with their children. During the first year of adoption, parents decided to share only some information and details about their children’s past experiences as they were worried about emotionally hurting them, with a higher fear of disclosing information about their children’s biological mothers. Although not shared in the first year, these details and information were communicated by parents in the following years, when parents considered the children aged enough to understand all aspects of their history cognitively and emotionally.

As for difficulties experienced while openly communicating in the first year, the most frequent fatigue was understanding children’s emotions during the CO, followed by personal fatigue by the parents in controlling their emotions. We did not have specific hypotheses regarding the communicative characteristics during the first year. Nevertheless, we believe that these results could be in line with the previous studies highlighting that adoption-related communication could be challenging for adoptive parents [4,14,15]. The challenges and difficulties experienced mainly by parents are related to emotional comprehension and expression, an aspect that has been highlighted as a critical factor for a positive CO [2,14,16,17]. Openly communicating about adoption-related themes requires adoptive parents to not only disclose information, but also share the emotional aspects of adoption itself, being emotionally tuned to the child’s feelings and able to support the child’s needs of understanding and giving sense to his or her roots and past [17,26]. Although adoptive parents are asked to guide their children in handling adoption-related difficult emotions, the current study’s results showed that sometimes, parents struggle to deal with this task.

We also found that the difficulties experienced by parents are related to some parents’ characteristics, such as avoidant attachment, perspective-taking abilities, and personal distress. Our results show that avoidant attachment and empathic concerns when exposed to children’s negative experiences could be risk factors during CO. On the other hand, adequate perspective-taking abilities could help parents share feelings during CO. We did not find any significant association between parents’ attachment anxiety and difficulties during CO. These results of the role of attachment avoidance, perspective-taking abilities, and personal distress align with previous studies [33], indicating that parents showing more empathy toward their children’s needs seem to support them in exploring their personal roots and origins. As for the attachment, our results emphasized a connection between avoidant attachment and parental difficulties in sharing personal feelings during CO. These results align with research stating that adult attachment orientations are associated with how people communicate with significant others [41,42]. Different attachment models can be associated with specific interpersonal emotion regulation strategies, ultimately influencing communication patterns between family members. Communicative processes such as self-disclosure, concealing information, and affectionate communication may vary, reflecting individual differences in the levels of attachment avoidance and anxiety [43,44,45]. For instance, a secure attachment style could foster interpersonal closeness behaviors, while attachment anxiety could promote excessive or inappropriate self-disclosure [43]. Finally, an avoidant attachment could be related to withholding information from significant others [44,45]. Consistently, our findings showed that avoidant attachment, usually related to discomfort with emotional closeness and interpersonal intimacy in close relationships [38], was associated with more difficulties communicating about adoption.

The current findings advance the literature regarding adoption-related communication at the beginning of adoptive family life by exploring how parents’ characteristics may be related to it. Indeed, although considerable studies regarding open communication, its development during the family life cycle, and effects on children have been conducted, little research has focused on the role of some specific parental characteristics in the open communication process. The novel aspect of the current study lies in exploring parental attachment and empathy as connected with communicative openness during the first year. Moreover, the present study reports parental experiences during the first year of adoption, aiming to collect parents’ emotions, memories, perceptions, and considerations about open communication at the beginning of their relationships with their children.

The current study could be helpful for researchers and clinicians working in the adoption field. Indeed, practitioners aware of parents’ experiences during the first year of adoption could construct preventive and post-adoption interventions or guide the adoptive families with post-adoption services in a more aware way.

However, our study presents some limitations that should be noted. First, we have relied only on retrospective memories and reports of adoptive parents and did not measure CO directly during the first year of adoption. This aspect should be taken into consideration when interpreting the results, as most of the adoptive parents who participated in the current study adopted their children several years before. Therefore, parents’ memories could be partially biased by the time that has elapsed between the first year of adoption and current data collection. Furthermore, parents’ memories and retrospective narrations of their experience in the first year after adoption may have changed over the years, in line with the development of both the children and the family. Moreover, memories could be influenced by parents’ current relationships with their children and by social desirability. The time distance between the first year of adoption and the data collection for the current study should be considered when interpreting the results. Second, we did not use a standardized instrument to evaluate CO in the first year, but we created an ad hoc questionnaire focused on the actual memories of experiences related to the first year after adoption. This aspect could influence our results. Third, our sample was primarily adoptive mothers, with a small percentage of fathers. Fourth, we collected information only from adoptive parents, without interviewing adopted children and other family members (e.g., siblings and grandparents). Fifth, the data were correlational, so they did not permit us to draw causal conclusions, and our models showed limited predictive power. Finally, our sample was not nationally representative of the Italian context, as most participants were from Northern Italy and had a high level of education.

Future directions could be to collect data from a larger sample, involving more adoptive fathers and other significant adults for adopted children and being more representative of the Italian context. Moreover, it would be interesting to conduct in-depth interviews to increase our knowledge about parental experiences while openly communicating during the first year of adoption, and to explore this issue by directly interviewing the adopted children.

## 5. Conclusions

The first year of adoption is particularly challenging for adoptive families as parents and children need to know each other, construct a family identity, develop a common narrative, and create an attachment bond. Open adoption-related communication (CO) is essential for all these psychologically intense processes. Nevertheless, a gap in the literature is still present about what happens during the first year of adoption and whether CO during this year could be linked to parents’ characteristics.

We examined open communication about adoption in the first year of adoption, retrospectively investigating adoptive parents’ experiences and difficulties, and the role of parents’ attachment and empathy in first year’s communication.

Our findings allow us to draw some relevant conclusions. First, we found that parents were worried about emotionally hurting their children, and, due to this, they communicated only some information about their children’s past experiences. Moreover, parents reported difficulties understanding children’s emotions and controlling personal emotions during CO. Second, we found that parents with an avoidant attachment and personal distress in empathy were more likely to feel fatigued in sharing and controlling personal feelings and understanding the feelings of their children, while perspective taking was a positive factor for the CO.

The knowledge developed and the concepts presented in this study can help researchers and clinical psychologists work with adoptive families.

## Figures and Tables

**Table 1 healthcare-11-03128-t001:** Difficulties experienced by parents.

	Yes		No	
	N	%	N	%
Difficulty in sharing personal feelings	94	34	185	66
Fatigue in controlling personal emotions	124	44	155	56
Difficulty in understanding children’s emotions	150	55	123	45
Fatigue in handling children’s questions	96	35	179	65

**Table 2 healthcare-11-03128-t002:** Parents’ attachment and empathy.

	N	M	SD	Min.	Max.
Avoidance	275	50.61	12.87	23.00	92.00
Anxiety	275	48.06	16.23	20.00	112.00
Perspective taking	280	2.71	1.27	1.29	18.00
Empathic concern	278	3.46	1.73	2.14	26.00
Personal distress	278	1.69	1.10	0.29	14.00

Legend: avoidance and anxiety are attachment dimensions evaluated using the ECR-R, while perspective-taking, empathic concern and personal distress refer to empathy dimensions measured with the IRI.

**Table 3 healthcare-11-03128-t003:** Point-biserial correlations between parental attachment, empathy, and difficulties experienced in the first year.

	Avoidance	Anxiety	PT	EC	PD
Sharing feelings	0.195 **	0.049	−0.051	0.027	0.175 **
Controlling emotions	0.082	0.039	−0.049	0.030	0.146 *
Understanding emotions	0.136 *	0.105	−0.049	0.017	0.116
Handling questions	0.059	−0.036	−0.022	0.054	0.171 **

The first column reports the difficulties in sharing personal feelings, controlling personal emotions, understanding children’s emotions, and handling children’s questions. Avoidance and anxiety are the two dimensions of attachment (measured with the ECR-R). PT: perspective taking, EC: empathic concern, PD: personal distress (measured via the IRI); * *p* < 0.05; ** *p* < 0.005.

**Table 4 healthcare-11-03128-t004:** Binomial logistic regressions (N = 290).

	Sharing Feelings				Controlling Emotions			
	β (SE)	Z	OR (95% CI)	*p*	B (SE)	Z	OR (95% CI)	*p*
Avoidance	0.03 (0.01)	2.95	1.03 (1.01, 1.06)	0.003	0.01 (0.01)	0.98	1.01 (0.98, 1.03)	0.325
Anxiety	0.00 (0.00)	0.52	1.00 (0.98, 1.02)	0.59	−0.00 (0.00)	−0.17	0.99 (0.98, 1.02)	0.862
Perspective Taking	−0.46 (0.22)	−2.09	0.63 (0.40, 0.97)	0.036	−0.33 (0.19)	−1.68	0.71 (0.48, 1.06)	0.092
Empathic Concern	−0.08 (0.23)	−0.36	0.91 (0.58, 1.4)	0.71	−0.15 (0.21)	−0.75	0.85 (0.56, 1.29)	0.452
Personal Distress	0.78 (0.28)	2.77	2.20 (1.26, 3.84)	0.006	0.73 (0.25)	2.87	2.09 (1.26, 3.46)	0.004
	**Understanding emotions**				**Handling questions**			
	**β (SE)**	**Z**	**OR (95% CI)**	** *p* **	**B (SE)**	**Z**	**OR (95% CI)**	** *p* **
Avoidance	0.02 (0.01)	1.77	1.02 (0.99, 1.04)	0.077	0.02 (0.01)	1.88	1.02 (0.99, 1.05)	0.060
Anxiety	0.01 (0.00)	1.38	1.01 (0.99, 1.03)	0.166	−0.00 (0.00)	−0.80	0.99 (0.97, 1.01)	0.423
Perspective Taking	−0.29 (0.22)	−1.30	0.74 (0.48, 1.15)	0.193	−0.38 (0.21)	−1.81	0.67 (0.44, 1.03)	0.070
Empathic Concern	0.07 (0.26)	0.26	1.07 (0.63, 1.80)	0.792	−0.08 (0.23)	−0.38	0.91 (0.58, 1.44)	0.698
Personal Distress	0.43 (0.25)	1.72	1.55 (0.94, 2.54)	0.084	0.69 (0.27)	2.52	1.95 (1.16, 3.41)	0.012

Legend: avoidance and anxiety are attachment dimensions evaluated using the ECR-R, while perspective-taking, empathic concern, and personal distress refer to empathy dimensions measured via the IRI.

**Table 5 healthcare-11-03128-t005:** Goodness of fit.

	Sharing Feelings	Controlling Emotions	Understanding Emotions	Handling Questions
X2 (df)	29.7 (7) **	16.1 (7) *	27.7 (7) **	17.3 (7) *
R2 McF	0.095	0.048	0.082	0.055
R2N	0.157	0.085	0.143	0.095

* *p* < 0.05; ** *p* < 0.001.

## Data Availability

Data are contained within the article.

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
