# Peer review of "A Retrospective Study on Adoptive Parenthood in the First Year after the Adoption: The Role of Parents’ Attachment and Empathy on Communicative Openness"

_healthcare, 2023, doi:10.3390/healthcare11243128_

Round 1

Reviewer 1 Report

Comments and Suggestions for Authors

Please see the attachment, thank you

Author Response

We thank you for the revision. Please find attached a Word document providing all our responses.

Reviewer 2 Report

Comments and Suggestions for Authors

The paper entitled “A retrospective study on adoptive parenthood in the first year after the adoption: the role of parental attachment and empathy on communicative openness” is well organized and written. The authors presented a detailed literature review of past studies on the analyzed issue, including a presentation of the relationship between studied parents’ characteristics. According to the authors “the interplay between parental empathy  and open communication regarding adoption has not been widely investigated yet” (p.4). By saying so, the authors point to the originality of the conducted research, but it raises some doubts because the association between empathy and communication is rather widely examined in the literature on the subject, including the issues of the functioning of adoptive families e.g.Faver & Alains (2012); Stone (2015) and many more.

Moreover, one can say that by measuring the level of empathy we can say little about parental empathy, as these two constructs although interrelated are not directly similar (for review see Stern et al., 2015 or Gonzales, 2021). By adapting IRI scale the authors seem to share this opinion, although there is nothing about it in the introduction section. Similarly, the suggested gap in the first year’s communication about adoption and psychological characteristics of parents is questionable. For example, such analysis has been recently conducted by Santona et al., 2022 – one of the authors of this paper. The presented study procedure lacks information about the duration and the date of data collection. In the abstract section, the authors suggested that they had interviewed participants, however, the online data collection method makes this sentence untrue. The adapted IRI scale despite reliability lacks information on structure – the authors simply assumed that it is similar to in original version, however at least EFA should be reported, especially as EC reliability is rather low, and I can not agree with the conclusion that IRI subscales show “adequate reliability”. Also, the scores from the CO scale created by the authors are rather poorly described, there are no psychometric properties of this measurement at all. There is no provided information on how the results were calculated, e.g. who decided about categories of difficulties, based on what they were divided, etc. The data analysis must be supplemented with sample size requirements for each calculated statistic. According to the authors, they performed logistic regression “to test all the hypotheses”. This is inconsistent with the three hypotheses posed by researchers, as each of them refers to comparison tests, and there is no study hypothesis about prediction.

In the result section the authors used t Student statistic to calculate gender differences, however, the predomination of females suggests that the parametric test is incorrectly used.

The authors divided participants into two groups – however, did not present information about the frequency of these two subsamples.

The goodness of fit for models should be discussed in the result section.

In the discussion section, the authors used the following sentence: “The current findings advance the literature regarding adoption-related communication at the beginning of adoptive family life by exploring how parental characteristics influence it.”, forgetting that in cross-sectional studies any causal conclusion can not be made. The word “influence” suggests causal effects.

In my opinion, the main disadvantage of this study is the time duration from the first year of adoption and this must be more discussed in the study limitation section. As a matter of fact, from the information provided by the authors “Adoptive parents and adopted children have lived together meanly for 10.2 years (SD = 5.18).” – this is crucial for the results. The title directly suggests the retrospective nature of the analysis, however, little is written on biases in memories and social approval susceptibility of participants caused by such study design. For example, the present relationship with adopted children may lead to overestimating or underestimating experienced difficulties.

In the conclusion section, some sentences should be rewritten as some stylistic errors are made e.g. lines 500-1.

All abbreviations must be explained below tables or when first used.

Comments on the Quality of English Language

The quality of english language is correct, minor revision is required.

Author Response

We thank you for your revision. Please find attached a Word document providing all our responses to your comments and suggestions. 

Round 2

Reviewer 2 Report

Comments and Suggestions for Authors

The authors made all suggested corrections, hence the paper may be published.